# Utility of Physiologically Based Pharmacokinetic Modeling to Investigate the Impact of Physiological Changes of Pregnancy and Cancer on Oncology Drug Pharmacokinetics

**DOI:** 10.3390/pharmaceutics15122727

**Published:** 2023-12-04

**Authors:** Xinxin Yang, Manuela Grimstein, Michelle Pressly, Elimika Pfuma Fletcher, Stacy Shord, Ruby Leong

**Affiliations:** Office of Clinical Pharmacology, Center for Drug Evaluation and Research, U.S. Food and Drug Administration, 10903 New Hampshire Avenue, Silver Spring, MD 20993, USA; xinxin.yang@bms.com (X.Y.); manuela.grimstein@fda.hhs.gov (M.G.); stacy.shord@fda.hhs.gov (S.S.)

**Keywords:** physiologically based pharmacokinetic (PBPK), model-informed pharmacokinetics, pregnancy, cancer, paclitaxel, docetaxel, acalabrutinib

## Abstract

Background: The treatment of cancer during pregnancy remains challenging with knowledge gaps in drug dosage, safety, and efficacy due to the under-representation of this population in clinical trials. Our aim was to investigate physiological changes reported in both pregnancy and cancer populations into a PBPK modeling framework that allows for a more accurate estimation of PK changes in pregnant patients with cancer. Methods: Paclitaxel and docetaxel were selected to validate a population model using clinical data from pregnant patients with cancer. The validated population model was subsequently used to predict the PK of acalabrutinib in pregnant patients with cancer. Results: The Simcyp pregnancy population model reasonably predicted the PK of docetaxel in pregnant patients with cancer, while a modified model that included a 2.5-fold increase in CYP2C8 abundance, consistent with the increased expression during pregnancy, was needed to reasonably predict the PK of paclitaxel in pregnant patients with cancer. Changes in protein binding levels of patients with cancer had a minimal impact on the predicted clearance of paclitaxel and docetaxel. PBPK modeling predicted approximately 60% lower AUC and C_max_ for acalabrutinib in pregnant versus non-pregnant patients with cancer. Conclusions: Our results suggest that PBPK modeling is a promising approach to investigate the effects of pregnancy and cancer on the PK of oncology drugs and potentially inform dosing for pregnant patients with cancer. Further evaluation and refinement of the population model are needed for pregnant patients with cancer with additional compounds and clinical PK data.

## 1. Introduction

Multiple population-based studies conducted in Australia, Europe, Canada, and the United States have reported a notable rise in cancer cases during pregnancy. The estimated incidence rates range from 89 to 191 per 100,000 pregnancies [1,2,3,4]. Cancers reported in pregnancy include breast cancer, melanoma, thyroid cancer, cervical cancer, lymphomas, and leukemias, in the order of decreasing frequency [5]. This increase in reported cancer cases associated with pregnancy can potentially be attributed to delayed maternal age, advancements in diagnostic technologies, and increased healthcare encounters during pregnancy [1].

Cancers presenting during pregnancy pose a growing public health concern, posing a significant challenge in terms of medical management. Patients with their health care providers face challenges in assessing the benefits vs. risks to the mother and fetus of use and timing of cancer treatment during various stages of pregnancy. These considerations may differ for an individual based on their cancer, stage of pregnancy, and available treatments. Many healthcare providers exhibit hesitancy in utilizing chemotherapy due to uncertainties about the exposure effect of cytotoxic drugs on the developing fetus. Moreover, there is a lack of information to guide healthcare providers regarding the appropriate dosages for molecularly targeted oncology drugs during pregnancy. These knowledge gaps regarding the safety and efficacy of oncology drugs during pregnancy are partially due to the initial exclusion of pregnant patients from cancer clinical trials and practical obstacles such as inadequate enrollment numbers, challenges in sample collection, and difficulties in follow-up procedures, etc.

Due to the limited availability of clinical pharmacokinetics (PK) data in pregnant patients with cancer, there is a growing interest in utilizing physiologically based pharmacokinetics (PBPK) modeling as a potential approach to predict PK changes for these drugs within this unique population [6]. It is widely recognized that the physiological changes occurring during pregnancy can significantly influence the absorption, distribution, metabolism, excretion (ADME), and PK of drugs [7], and these changes have been incorporated into pregnancy PBPK models [8,9,10,11]. However, physiological changes have also been reported in patients with various cancer types that would need to be considered in modeling efforts related to cancer treatment in pregnancy. For example, multiple studies have consistently reported a lower level of plasma albumin and an increased level of alpha-1-acid glycoprotein (AGP) in various cancer types [12,13,14,15]. In addition, there is growing evidence suggesting alterations in the expression of cytochrome P450 (CYP) enzymes in patients with cancer [12,16,17]. Moreover, renal function tends to decrease in patients with cancer [18]. In contrast to the changes reported in patients with cancer, both plasma albumin and AGP levels tend to decrease throughout pregnancy [8,9,19,20]. Studies have reported a large variation of increased activity of CYP3A4 during pregnancy [8,21], and in vitro studies demonstrated that pregnancy-related hormones (PRH) significantly induce CYP2C8 protein levels [22,23]. Lastly, a meta-analysis showed an approximately 35% increase in glomerular filtration rate (GFR) throughout pregnancy [8]. An overview of the key studies and their findings regarding the physiological changes in pregnant and cancer populations compared to healthy subjects is listed in Table 1.

To the best of our knowledge, PBPK modeling efforts have primarily focused on gathering and refining anatomical and physiological parameters to construct separate virtual populations representing pregnancy [8,9,11] or cancer conditions [13] but not both populations combined. While these efforts provide a valuable foundation, we specifically investigated the combined impact of pregnancy and cancer on the PK of oncology drugs.

Therefore, our research aims to investigate physiological changes reported in both pregnancy and cancer populations into a PBPK modeling framework that allows for a more accurate estimation of PK changes in pregnant patients with cancer.

## 2. Materials and Methods

### 2.1. Compound Selection and Clinical Data Extraction

Paclitaxel and docetaxel were chosen as probe compounds for our study due to the availability of clinical PK data in pregnant patients with cancer. These compounds were used to assess the impact of physiological changes associated with pregnancy on the PK of pregnant patients with cancer. Additionally, acalabrutinib was chosen for PBPK model application, given that one of the animal preclinical studies indicated no embryo-fetal toxicity or mortality and that it has an approved indication for cancers common in pregnancy [27].

Relevant data, including dosing regimens, demographic information, pertinent PK parameters, and drug concentration–time profiles, were obtained for paclitaxel, docetaxel, and acalabrutinib. Detailed descriptions of model development and verification for each drug and associated datasets are provided in the Appendix A. The general workflow of the PBPK modeling strategy is described in detail below and represented in Figure 1.

### 2.2. PBPK Model Development and Application

The overall PBPK modeling strategy consisted of two primary steps: the development and verification of individual drug models, and the establishment of a population model representing pregnant patients with cancer.

In the first step, PBPK drug models for paclitaxel, docetaxel, and acalabrutinib were developed and verified using multiple sets of clinical PK data obtained from healthy subjects and/or non-pregnant patients with cancer.

In the second step, a population model representing a healthy subject was modified by incorporating the physiological changes induced by pregnancy. The resulting pregnancy population model was then verified against relevant clinical PK data of paclitaxel and docetaxel obtained from pregnant patients with cancer, using the previously validated paclitaxel and docetaxel drug models. The effect of physiological changes associated with cancer was then evaluated to determine if including these changes in the PBPK model would improve the prediction compared to the pregnancy model. The aim of this verification was to demonstrate that the model adequately predicts the reported PK of docetaxel and paclitaxel in pregnant patients with cancer.

A verified population model representing pregnant patients with cancer was then applied to predict the PK for acalabrutinib.

### 2.3. Drug Model Development and Verification

The PBPK model of paclitaxel and acalabrutinib was adopted by Mendes et al. [28] and Zhou et al. [29]. New models were developed for docetaxel. The key ADME characteristics of each drug are listed below. The physicochemical properties and in vitro and clinical data used to build the models can be found in the Appendix A.

Briefly, paclitaxel is primarily metabolized by CYP2C8 and CYP3A4 (80% of total clearance), with minor contributions of renal elimination and biliary excretion mediated by P-glycoprotein (P-gp). Paclitaxel is significantly bound to plasma protein (mainly albumin), with a fraction unbound in plasma (fu) of approximately 0.05. It exhibits non-linear PK over the dose range of 60–200 mg/m^2^, with dose-dependent effects on paclitaxel plasma protein binding, volume of distribution, and drug clearance [28].

Docetaxel is significantly bound to plasma proteins (albumin, AGP, and lipoprotein) with a fu of approximately 0.06 in cancer patients. The PK appears to be linear for doses between 50 and 200 mg/m^2^. Metabolism by CYP3A4/5 accounts for around 80% of the total clearance of docetaxel, with minor contributions of renal and biliary clearances [30,31].

Acalabrutinib is significantly bound to plasma protein with a fu of 0.03 and exhibits almost linear PK across the dose range of 75 to 250 mg [32]. It is predominantly metabolized by CYP3A4 (around 80% of total clearance), with a lesser contribution from glutathione conjugation and amide hydrolysis [29].

Clinical PK data from healthy subjects, patients with cancer, and drug–drug interactions were utilized to develop and verify each drug model. PBPK simulations were performed using virtual subjects aiming to closely match the age and sex characteristics of the patients in the corresponding clinical studies. Detailed information regarding trial simulation design is provided in the Appendix A.

### 2.4. Physiological Changes Implemented in Cancer and Pregnancy Population Models

The physiological changes associated with cancer and pregnancy can lead to modifications in the ADME processes of drugs. In this study, we focused on evaluating the impact of cancer and pregnancy-associated changes on the levels of albumin and AGP, CYP3A4 and CYP2C8 expression and renal function, and effects on the PK of paclitaxel, docetaxel, and acalabrutinib (Table 2).

#### 2.4.1. Cancer Population Model

In the cancer population model (Simcyp V21), a 15% decrease in plasma albumin level and a 100% increase in AGP levels compared to healthy subjects is incorporated [13]. Regarding expression or activity of CYP enzymes in patients with cancer, the cancer model does not incorporate any changes in the expression of gut or liver CYP enzymes relative to a healthy subject. Likewise, abundance of the efflux transporter P-gp is similar between healthy subjects and patients with cancer. In addition, patients with cancer may present renal dysfunction/insufficiency with a lower GFR, which is represented by 30–40% decrease in GFR compared to healthy subjects in the cancer population model [33].

#### 2.4.2. Pregnancy Population Model

In the pregnancy model (Simcyp V21), plasma levels of albumin and AGP tend to decrease throughout the pregnancy, with around 25% and 30% less at 32 gestational weeks compared to pre-pregnancy levels, respectively [8]. In addition, a 25% increase in GFR throughout pregnancy is added to this model [8]. Further, the expression of CYP3A4 increases up to 130% of the pre-pregnancy level at the end of the third trimester [11], while no alteration of CYP2C8 expression is implemented [8]. We further explored an alternative CYP2C8 expression during pregnancy based on in vitro data [22,23]. Khatri et al. [23] reported a significant (150%) induction of CYP2C8 protein by PRH (such as progesterone and 17-β-estradiol) in vitro. This observation was consistent with a previous in vitro study [22] reporting that progesterone enhanced CYP2C8 mRNA levels by a magnitude comparable to the one observed with CYP3A4. A modified pregnancy model with an alternative CYP2C8 expression (150% increase) was evaluated in our PBPK analysis.

Simulations were also conducted using this modified pregnancy model with an alternative CYP2C8 expression, and results were compared with the clinical PK data in pregnant patients with cancer.

### 2.5. Development and Application of a Population Model for Pregnant Patients with Cancer

PBPK analysis and clinical PK data in non-pregnant or pregnant patients with cancer, treated with paclitaxel or docetaxel, were used to evaluate the impact of physiological changes of pregnancy and cancer on drug PK. The overall workflow for the development of a population model for pregnant patients with cancer is illustrated in Figure 2.

To begin, the predictive performance of both healthy subject and cancer population models was evaluated via comparison of predicted versus observed PK profiles of nonpregnant patients with cancer. This analysis sought to determine which population model better predicted the PK of pregnant patients with cancer and to identify any key physiological changes associated with cancer that should be incorporated into a population model representing pregnant patients with cancer.

Sensitivity analysis was also conducted to evaluate the impact on PK of key factors such as CYP abundance, binding proteins, and other clearance pathways. This analysis aimed to quantify the influence of these factors and their contribution to changes in drug PK during pregnancy and cancer.

Following this decision tree (Figure 2), the relevant pregnancy population model was then used to predict the PK in pregnant patients with cancer administered acalabrutinib. The virtual trial design was set to 10 trials of 10 patients with age range from 18 to 45 years at 32 gestational weeks. Dose and schedule of acalabrutinib reflect the current approved recommended dosage in non-pregnant patients with cancer. The simulated exposures were compared between non-pregnant patients with cancer and pregnant patients with cancer to assess any differences in drug exposure between these two populations.

### 2.6. Predictive Performance

The predictive performance among different population models in describing a drug PK profile was evaluated via visual comparison of predicted and observed plasma concentration-time profiles. The model was determined to adequately describe the PK profile when the observed data were within the 90% prediction interval (5th–95th percentile range of the virtual population). In addition, model predictions were evaluated via comparison of predicted PK parameters to reported values. The predicted to observed ratios of key pharmacokinetic parameters, such as maximum plasma concentration (C_max_), area under the concentration-time curve (AUC), and clearance (CL), were estimated. The model was acceptable when the predicted value fell within a two-fold range of the observed value, as a commonly used metric assessing PBPK model performance [34].

### 2.7. Software

The population-based PBPK software Simcyp^®^ (Version 21, Certara UK Limited, Sheffield, UK) was used as the platform for PBPK analyses. Simulations were performed using the Simcyp^®^ default virtual population models representing healthy subjects (“Sim-Healthy Volunteers” and “Sim-NEurCaucasian”), healthy pregnant subjects (“Sim-pregnancy”), and patients with cancer (“Sim-Cancer”). A modified pregnancy population model was also developed and used for simulations, as previously described.

In cases where clinical PK data were not directly reported in the reference, they were extracted using the WebPlotDigitizer (version 4.6, https://apps.automeris.io/wpd/ (accessed on 2 March 2023). In addition, the pharmacokinetic parameters were calculated via noncompartmental analysis using the Phoenix WinNonlin software (version 8.3, Pharsight Corp., Mountain View, CA, USA).

## 3. Results

### 3.1. Model Evaluation

#### 3.1.1. Predictive PK Performance of Drug Models in Non-Pregnant Population

##### Paclitaxel

The predictive PK performance of the paclitaxel drug model (Appendix A) was evaluated via comparison with PK data from three clinical studies in non-pregnant patients with cancer, administered with paclitaxel 175 mg/m^2^ [24,35,36]. Overall, the model predictions were within ±30% of reported PK values in both studies (Appendix A). The model adequately described the PK profiles of paclitaxel at a dose of 175 mg/m^2^ in the non-pregnant cancer population. Representative predicted versus observed plasma concentration–time profiles after a single 3 h IV infusion of paclitaxel 175 mg/m^2^ are shown in Figure 3a.

##### Docetaxel

The predictive PK performance of the docetaxel drug model (Appendix A) was evaluated via comparison with PK data from two clinical studies of docetaxel at doses of 100 mg/m^2^ in non-pregnant patients with cancer [37,39]. The model predictions for C_max_, AUC, and CL were around 20% of the reported values in both studies (Appendix A). Representative predicted versus observed plasma concentration-time profiles after a single 1 h IV infusion of docetaxel 100 mg/m^2^ are shown in Figure 3b.

##### Acalabrutinib

The predictive PK performance of acalabrutinib drug model (Appendix A) was evaluated by comparing the PK data from a clinical study in non-pregnant patients with cancer administered acalabrutinib 100 mg once daily or twice daily (BID), 175 mg once daily, and 250 mg once daily. The predicted AUC value was around 7% of the observed value, while the C_max_ was underestimated (32% of the observed value). We noted that high variability was reported for the C_max_ values in patients treated with acalabrutinib 100 mg BID (CV% of 102) [38]. The underestimation of C_max_ was not predicted for the other acalabrutinib dosages besides 100 mg BID (Appendix A). The model reasonably described the steady-state PK profiles of acalabrutinib for various dosages, with the exception of C_max_ of acalabrutinib 100 mg BID. Representative predicted versus observed plasma concentration–time profile of acalabrutinib 100 mg BID is illustrated in Figure 3c.

#### 3.1.2. Evaluating the Effect of Physiological Changes in Cancer on the PK of Paclitaxel and Docetaxel

Using paclitaxel and docetaxel drug models, simulations were conducted utilizing the “healthy subject” and cancer population models. By comparing the predicted PK profiles in healthy subjects and cancer populations, we aimed to identify the key cancer-specific physiological alterations that had an impact on ADME processes.

Paclitaxel is primarily bound to albumin, with an average unbound fraction in plasma of 0.06. A 15% decrease in plasma albumin levels, as reported in cancer populations, has been implemented in the default cancer population model (Table 2). This reduction in albumin levels led to an estimated 17% increase in the unbound fraction of paclitaxel in plasma (Appendix A). However, simulations indicated that this change had a minor impact on the predicted clearance of paclitaxel (Figure 4a,b). Since paclitaxel is primarily metabolized by CYP3A4 and CYP2C8 with a moderate hepatic extraction ratio, minor changes in its plasma unbound fraction would have a minimal effect on clearance compared to hepatic blood flow. Similarly, the decreased renal function in the cancer population (30–40% decrease in GFR) did not significantly affect the total clearance of paclitaxel, as only around 10% of the drug is eliminated via the renal route. Figure 4a,b shows the comparison of observed versus predicted clearance of paclitaxel using both healthy and cancer population models.

In the case of docetaxel, it binds to multiple plasma proteins such as albumin, AGP, and lipoproteins with an average unbound fraction of approximately 0.06. To account for the altered physiology observed in patients with cancer, the default cancer population model included a 2-fold increase in the AGP levels and a 15% decrease in the albumin levels, while no change was observed in lipoproteins. These changes in binding protein levels led to a 15% decrease in the unbound fraction of docetaxel in patients with cancer (fu = 0.06) compared to healthy subjects (fu = 0.07). However, this difference in unbound fraction had a minor impact (<15% change) on the predicted clearance of docetaxel between the healthy and cancer population models (Figure 4c,d, Appendix A). This implies that alterations in the binding of docetaxel to AGP did not significantly affect its overall clearance. Similar to paclitaxel, the decreased renal function in the cancer population did not significantly affect docetaxel clearance since the renal route only contributes to around 5% of total elimination (Appendix A).

Based on the analysis of those two drug examples, changes in binding protein levels and decreased renal function in cancer patients had minimal impact on the predicted clearance of paclitaxel and docetaxel. Therefore, those changes were not considered necessary to be accounted for in the pregnancy population model. Further, the cancer population model assumes a similar CYP expression as in the healthy population model. Our analysis seems to support this assumption as the predicted clearance of paclitaxel (a CYP3A4 and CYP2C8 substrate) and docetaxel (a CYP3A substrate) was not significantly different between these population models. Thus, the next step was predicting paclitaxel and docetaxel exposure using the software’s pregnancy population model.

### 3.2. Evaluating the Effect of Physiological Changes in Pregnancy on the PK of Paclitaxel and Docetaxel

The pregnancy population model used in this PBPK study included pregnancy-induced time-dependent changes in both albumin and AGP levels, CYP3A4 expression, and renal function, while no change in the expression of CYP2C8 (Table 2).

In two clinical studies of pregnant patients with cancer who received paclitaxel (175 mg/m^2^ over 3 h) [40,41], paclitaxel exposure, in terms of AUC and C_max_, appeared to be around 45% lower in the second and third trimesters [35]. Using the pregnancy population model, PBPK predictions overestimate paclitaxel exposure during the second and third trimesters of pregnancy. The predicted mean C_max_, AUC, and CL were 2.6 μg/mL, 11.8 μg/mL·h, and 27.7 L/h, respectively, and the respective digitized values were C_max_ of 2.2 μg/mL, AUC of 8.7 μg/mL·h, and CL of 35.7 L/h, from the van Hasselt et al. (2014) study in pregnant patients with cancer (n = 5, median gestational age in weeks (GW) = 23) [40] (Appendix A). Likewise, in the second study conducted by Janssen et al. (2021), in pregnant patients with cancer (n = 20, median GW = 31), the predicted mean Cmax, AUC and CL were 2.5 μg/mL, 11 μg/mL·h and 30.3 L/h, respectively, and the respective digitized values were C_max_ of 3.3 μg/mL, AUC of 8 μg/mL·h, and CL of 39.2 L/h [41] (Appendix A).

Paclitaxel (175 mg/m^2^) is primarily eliminated via CYP3A4 and CYP2C8 metabolism (similar contribution) and to a minor extent via renal elimination. While the pregnancy model, implemented approximately a 100% increase in CYP3A4 expression and a 25% increase in renal function at 32 gestational weeks (Table 2), both simulation results showed that the pregnancy model overestimated paclitaxel exposure (around 40% higher) in pregnant patients with cancer (Appendix A). Therefore, other physiological parameters that could affect paclitaxel clearance, such as CYP2C8 expression during pregnancy, were investigated. Based on in vitro data indicating an increase in CYP2C8 expression mediated by PRH, a 2.5-fold increase in CYP2C8 abundance at 32 weeks gestation was evaluated [23]. This modification in the pregnancy population model improved the prediction accuracy of the elimination phase of paclitaxel, resulting in the predicted AUC and CL values within ±25% of the observed values (Figure 5, Appendix A). Using the pregnancy model with altered CYP2C8 expression, the predicted mean AUC and CL of paclitaxel were 9.8 μg/mL·h and 33.2 L/h, respectively, at 32 gestational weeks.

In two clinical studies, in which pregnant patients with cancer received docetaxel 100 mg/m^2^ over 1 h [7,42], docetaxel exposure, in terms of both AUC and C_max_, was slightly lower (around 30%) in pregnant compared to non-pregnant patients with cancer [41]. The pregnancy population model accurately predicted the reported PK of docetaxel during the third trimester of pregnancy (predictions were within ±25% of observed values) (Figure 6). The predicted mean C_max_, AUC, and CL were 1.9 μg/mL, 2.8 μg/mL·h, and 65 L/h, respectively, compared to the respective digitized values of C_max_ of 1.7 μg/mL, AUC of 2.4 μg/mL·h, and CL of 75L/h, in pregnant patients with cancer (n = 3, median GW = 32) from the study conducted by van Hasselt et al. [40] (Appendix A). Likewise, the predicted mean Cmax, AUC, and CL were 1.9 μg/mL, 2.8 μg/mL·h, and 65L/h, respectively, compared to the respective digitized values of Cmax of 1.7 μg/mL, AUC of 2.3 μg/mL·h, and CL of 79L/h, in pregnant cancer patients (n = 9, median GW = 32) from the study conducted by Janssen et al. [41] (Appendix A). The pregnancy model accounted for pregnancy-induced changes in CYP3A4 abundance, which plays a major role in docetaxel elimination (80% of total clearance).

The model-predicted exposure of paclitaxel and docetaxel in the pregnant cancer population was compared between patients with cancer and pregnant patients. In the case of paclitaxel, simulation revealed a 38% lower C_max_ (2.3 mg/L vs. 3.8 mg/L) and a 45% lower AUC (9.8 mg·h/L vs. 17.5 mg·h/L) in pregnant patients compared with non-pregnant patients with cancer, which is in good alignment with the observed data (45% lower C_max_ and 46% lower AUC). For docetaxel, the simulations showed that the exposure in pregnant patients was around 25% lower in terms of both C_max_ (1.9 mg/L vs. 2.5 mg/L) and AUC (2.8 mg·h/L vs. 3.5 mg·h/L) compared to the non-pregnant cancer population. These findings were consistent with the observed data, which demonstrated a 34% lower C_max_ and a 27% lower AUC in pregnant patients with cancer (Appendix A).

### 3.3. Model Application

In previous sections, we demonstrated that both healthy and cancer population models had similar PK predictive performance for paclitaxel and docetaxel in patients with cancer, indicating that physiological changes implemented in the cancer model had minimal impact on the prediction of the PK of these CYP3A4 substrates.

Likewise, the PK of acalabrutinib in non-pregnant patients with cancer, at an approved dose of 100 mg BID, was similarly predicted by both healthy and cancer population models (Table 3). Following our proposed decision tree (Figure 2), the default pregnancy model could then be used to predict the PK of acalabrutinib in pregnant patients with cancer.

Following the oral administration of acalabrutinib (100 mg BID), simulations showed that the steady-state exposure in pregnant patients (32 GW) is expected to be around 60% lower in terms of both C_max_ (251 ng/mL vs. 566 ng/mL) and AUC_0–24h_ (cumulative AUC over 2 dosing intervals, 662 ng·h/mL vs. 1719 ng·h/mL) compared with non-pregnant patients with cancer (Table 3).

## 4. Discussion

There is a lack of information on appropriate dosages of molecularly targeted oncology drugs during pregnancy with currently limited information on PK and pregnancy outcomes in pregnant patients with cancer receiving molecularly targeted oncology drugs. Registry studies may provide an additional source of data in the future [43]. PBPK is a model-informed approach that is useful to understand both the effects of pregnancy and cancer on drug PK to potentially inform the dosing of oncology drugs for pregnant patients with cancer.

## 5. Clinical Implications

The results of our analysis show a reasonable prediction of the pregnancy-related changes in the PK of paclitaxel and docetaxel in patients with cancer. The predicted decrease in systemic exposure in pregnant patients, as determined using the PBPK approach, aligns well with previous analyses conducted using population PK approaches [40,41]. It is important to note that significant interpatient variability has been observed in clinical studies on docetaxel and paclitaxel.

Based on a model-based meta-analysis of 29 monotherapy trials for paclitaxel dose–response in different cancer types, paclitaxel efficacy (e.g., objective response rate, median overall survival, and progression-free survival) is correlated with the average dose per week (mg/m^2^/week), and safety (e.g., incidence of neutropenia) is correlated with dose per administration (mg/m^2^) [44]. Lu et al. suggested that the paclitaxel dosage of 65–80 mg/m^2^ every week has comparable to better efficacy and lower neutropenia incidence compared to that of 175 mg/m^2^ every 3 weeks [44]. Collectively, the observed 46% lower paclitaxel AUC in pregnant patients with cancer and known dose–response and exposure–response relationships for the efficacy and safety of paclitaxel could be used to select a dosage in pregnant patients with cancer based on exposure matching to the dosages commonly administered to patients with cancer.

Based on a model-based meta-analysis of 46 monotherapy trials for docetaxel in patients with advanced non-small cell lung cancer (NSCLC), there were dose–response relationships for objective response rate and neutropenia [45]. In a randomized dose-finding trial of docetaxel 60, 75, or 100 mg/m^2^ Q3W in patients with advanced breast cancer, higher response rates and improved time to the progression along with higher incidences of most hematologic (e.g., Grade 3–4 neutropenia, febrile neutropenia) and nonhematologic toxicities were observed with increased doses; however, there was no statistically significant relationship between docetaxel exposure (AUC) and objective response rate [46]. Together, the observed 27% lower docetaxel AUC in pregnant patients with cancer and known dose–response and exposure–response relationships for the efficacy and safety of docetaxel may support selecting a dosage for pregnant patients with cancer.

It was reasonable to predict the exposure of acalabrutinib in pregnant patients with cancer using the pregnancy population model based on the following rationale: (1) the success of using the default pregnancy population model in predicting the exposure of docetaxel reported in pregnant patients with cancer, (2) the indication that changes in binding protein levels in cancer patients have minimal impact on the predicted clearance of paclitaxel and docetaxel, and (3) the similarities in ADME properties between acalabrutinib, paclitaxel, and docetaxel, especially pertaining to the major contribution of CYP3A enzyme to drug clearance and plasma protein binding affinity.

The simulations using the pregnancy population model predicted approximately 60% lower AUC_0–24h_ and C_max_ of acalabrutinib. There is no statistically significant relationship between acalabrutinib exposure (AUC_0–24h_ and C_max_) and efficacy (e.g., overall response rate) in patients with mantle cell lymphoma; however, a positive relationship was identified between the exposure (AUC_0–24h_) and the probability of ≥Grade 2 neutropenia. Increasing the dose from 100 mg to 200 mg twice daily would increase the probability of Grade 2+ neutropenia from 7.5% (95% CI: 4.9%, 11.2%) to 13% (95% CI: 7.9%, 20.8%). In addition, acalabrutinib at a dosage of 100 mg BID resulted in maximal target, Bruton tyrosine kinase (BTK), occupancy with the least inter-patient variability at steady-state trough and increasing the dosage higher than 100 mg BID did not appear to further increase BTK occupancy at steady-state trough and may result in off-target effects [32]. Similar exposure–response relationships were observed with data from patients with chronic lymphocytic leukemia (CLL) or small lymphocytic lymphoma (SLL) [42]. The simulated lower exposure and known dose–response and exposure–response relationships for the efficacy, safety, and pharmacodynamics of acalabrutinib may potentially inform the dosing of acalabrutinib for pregnant patients with cancer.

## 6. Limitations

One limitation of a PBPK model representing the cancer population is the uncertainty surrounding CYP enzyme expression in this population. The current (Simcyp^©^ V21) cancer population model assumes no alterations in CYP enzymes in patients with cancer compared to healthy subjects. However, the clinical effect of cancer on CYP-mediated drug clearance has not been established. For instance, CYP3A downregulation has been reported in patients with cancer during an acute inflammatory response [12]. This alteration (30% reduction in hepatic and intestinal CYP3A4 expression) has been investigated in a PBPK study demonstrating a better prediction of exposure to sensitive CYP3A4 substrates (namely, midazolam and simvastatin) in patients with cancer [16]. In contrast, Cheeti et al. [13] suggested that CYP3A activity is not altered in patients with cancer based on PBPK modeling of midazolam exposure. This finding is supported by Baker et al. [47], who examined CYP3A activity in 134 patients with cancer and found no significant changes related to age, sex, or body size. For CYP2C8, most studies reported minimum to no changes in patients with cancer [16,17]. The reader is referred to published reviews for further information on this topic [12,48]. While our analysis using the cancer population model did not suggest alterations in CYP3A4 and CYP2C8 expressions in patients with cancer, it is essential to consider the limited number of drugs evaluated and their ADME properties (e.g., the contribution of CYP3A4 to overall clearance being equal or less than 80%) and cancer type. Additional investigational data (e.g., in vitro, clinical, and model-based) are needed to reach a consensus on the effect of cancer on CYP-mediated drug clearance.

While it is recognized that CYP3A4 expression increased during pregnancy, there are knowledge gaps on the effect of pregnancy on other relevant enzymes, such as CYP2C8 which is involved in the metabolism of paclitaxel. The current (Simcyp© V21) pregnancy population model assumes no changes in CYP2C8. Supported with the available in vitro data [22,23], our PBPK analysis suggested a significant induction of CYP2C8 expression during pregnancy based on the clinical PK data of paclitaxel in pregnant patients with cancer. While further modeling analysis to verify the proposed hypothesis is warranted, this evaluation may be limited by the availability of clinical PK data of sensitive CYP2C8 substrates from healthy pregnant populations.

Another limitation of our study is that the predictive performance of the Simcyp^©^ (V21) pregnancy population model and modified pregnancy population model were only evaluated using paclitaxel and docetaxel. Therefore, the applicability of the PBPK model to predict the exposure of molecularly targeted oncology drugs may be limited. It is crucial to consider the similarity in ADME properties between the drugs of interest and paclitaxel/docetaxel before making predictions using the model. Furthermore, to establish a robust population model representing pregnant patients with cancer, more comprehensive information is needed to fill the current knowledge gaps. This includes obtaining data on pregnancy and/or cancer-dependent changes in the physiological parameters, such as protein binding, enzyme activity, and renal function. Additional model evaluation with other drugs with diverse ADME properties and information regarding protein binding, CYP expression, and clinical PK data in this population is necessary to continually improve our understanding of the effect of intrinsic factors, such as pregnancy and cancer, on drug PK.

## 7. Conclusions

PBPK modeling could be used to predict the PK changes in pregnant patients with cancer given the limited availability of observed clinical PK data in this specific population. The Simcyp pregnancy population model reasonably predicted the PK of docetaxel in pregnant patients with cancer, while the modified pregnancy population model, including a pregnancy-induced increase in CYP2C8 abundance, improved the prediction accuracy of paclitaxel PK in pregnant patients with cancer. Considering the similarities in ADME properties between acalabrutinib and docetaxel, it is possible, within reasonable error, to predict the exposures of acalabrutinib in pregnant patients with cancer using the default pregnancy population model. Further studies of acalabrutinib in pregnant patients with cancer and clinical PK data are needed for the verification of the PBPK predictions.

Our results suggest that pregnancy PBPK modeling is a promising approach for investigating both the effects of pregnancy and cancer on drug PK to potentially inform the dosing of oncology drugs for pregnant patients with cancer. Further evaluation and refinement of the population model for pregnant patients with cancer with additional compounds and clinical PK data are needed.

## Figures and Tables

**Figure 1 pharmaceutics-15-02727-f001:**
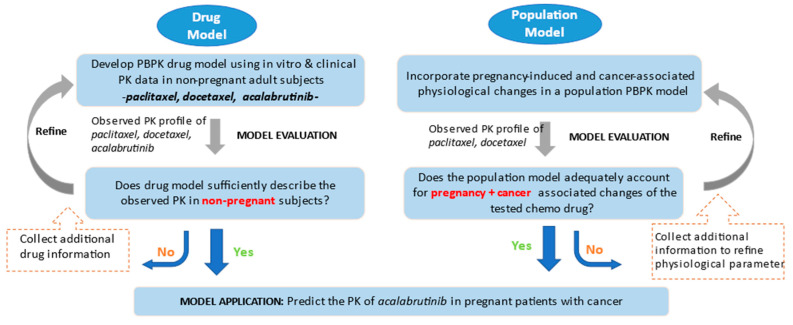
Workflow for the PBPK modeling strategy.

**Figure 2 pharmaceutics-15-02727-f002:**
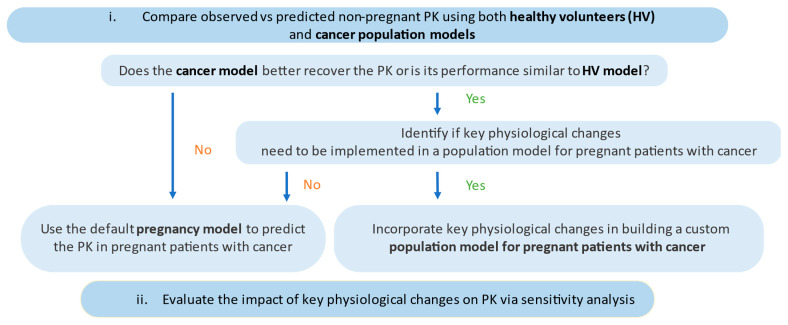
Workflow of the development process of a population model for pregnant patients with cancer.

**Figure 3 pharmaceutics-15-02727-f003:**
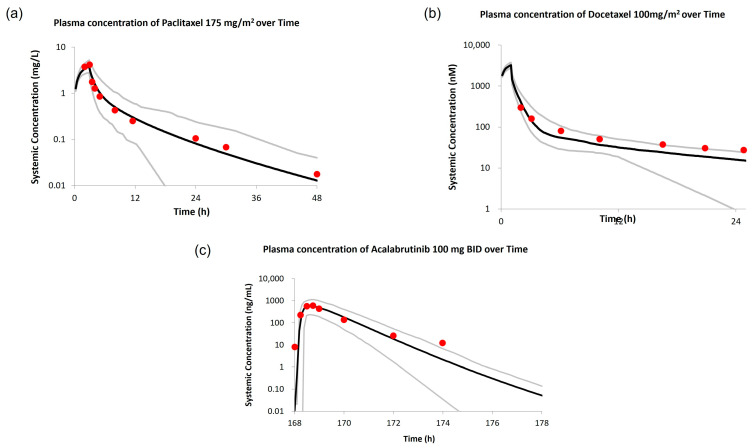
Comparison of PBPK predicted versus observed plasma concentration–time profiles in non-pregnant patients with cancer. (**a**) Paclitaxel (3 h infusion of 175 mg/m^2^). Red circles are mean observed PK [35]. (**b**) Docetaxel (1 h infusion of 100 mg/m^2^). Red circles are median observed PK (cite reference) [37]. (**c**) Acalabrutinib (100 mg BID on day 8). Red circles are mean observed PK [38]. Black and grey lines represent the population and 5th–95th percentiles of the predicted mean PK profiles, respectively.

**Figure 4 pharmaceutics-15-02727-f004:**
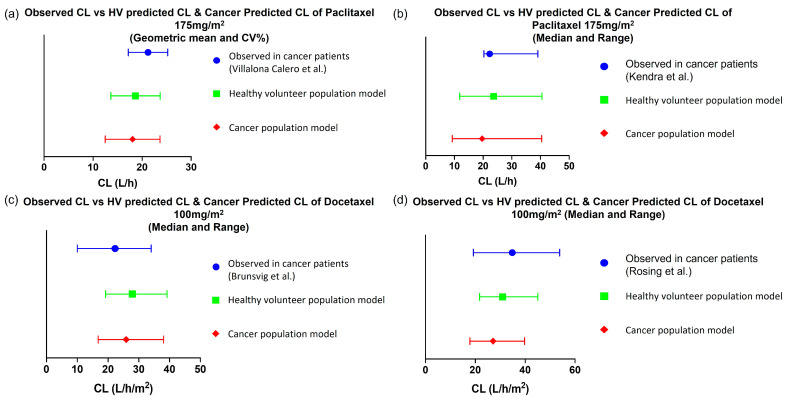
Comparison of observed clearance versus PBPK predictions using the default healthy volunteers or cancer population models. (**a**) Paclitaxel observed clearance (Villalona-Calero et al. 1999) [35] after single administration of 175 mg/m^2^ over 3 h. (**b**) Paclitaxel observed clearance (Kendra et al. 2015) [24] after single administration of 175 mg/m^2^ over 3 h. (**c**) Docetaxel observed clearance (Brunsvig et al. 2007) [37] after single administration of 100 mg/m^2^ over 1 h. (**d**) Docetaxel observed clearance (Rosing et al. 2000) [39] after single administration of 100 mg/m^2^ over 1 h.

**Figure 5 pharmaceutics-15-02727-f005:**
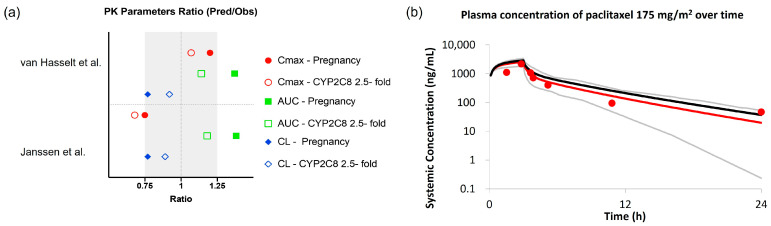
Comparison of observed paclitaxel PK versus PBPK predictions using the default or modified pregnancy population models. (**a**) Ratios are expressed as model-predicted values of PK parameters over observed values from van Hasselt et al. [40] and Janssen et al. [41] studies, for C_max_ (red circle), AUC (blue square), and CL (green diamond). The solid symbols represent the results using the default pregnancy population model, while the open symbols represent the results using a modified pregnancy model with a 2.5-fold increase in CYP2C8 abundance at 32 gestational weeks. Grey rectangle indicated ±25% prediction error; (**b**) PBPK predicted versus observed plasma concentration–time profiles of paclitaxel in pregnant patients with cancer after 3 h infusion of 175 mg/m^2^, using the default and modified pregnancy model. Red circles represent observations, red and grey lines represent the mean and 5th–95th percentiles of the predicted PK profile using the modified pregnancy model; the black line represents the predicted mean PK profile using the default pregnancy model.

**Figure 6 pharmaceutics-15-02727-f006:**
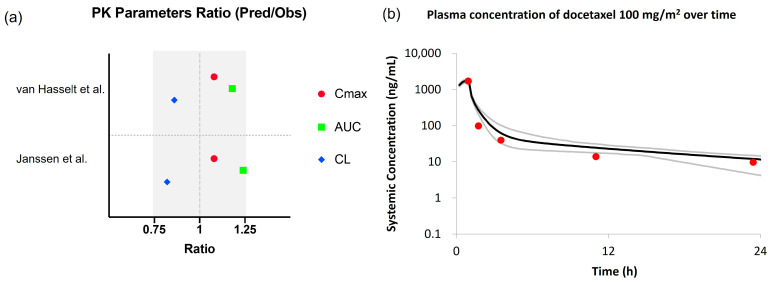
Comparison of observed docetaxel PK versus PBPK predictions using the default pregnancy population model. (**a**) Ratios are expressed as model-predicted values of PK parameters over observed values from van Hasselt et al. [40] and Janssen et al. [41] studies, for C_max_ (red circle), AUC (blue square), and CL (green diamond). The solid symbols represent the results using the default pregnancy population model. Grey rectangle indicated ±25% prediction error; (**b**) PBPK predicted versus observed plasma concentration-time profiles of docetaxel in pregnant cancer patients after 1 h infusion of 100 mg/m^2^, using the default pregnancy model. Red circles represent observations, black and grey lines represent the population and 5th–95th percentiles of the predicted mean PK profile.

**Table 1 pharmaceutics-15-02727-t001:** Summary of key physiological changes reported in pregnant and cancer populations compared with healthy subjects.

	Healthy Subjects	Cancer	Pregnancy
Albumin (g/L)	40–50 [12]	35–41 [12,13,14]	38.5 (30 GW), 37.6 (34 GW) [8]
Alpha-1-acid glycoprotein (AGP) (g/L)	0.5–1 [15]	1.34–1.38 [12,13,14]	0.6 (30–35 GW) [8]
CYP3A4 expression/activity	--	10–33% decrease [12,16,17], No change [13,24]	60% increase throughout [25], 75% increase (27 GW), 130% increase (at term) [11], 50–100% increase [21]
CYP2C8 expression/activity	--	No change to minimum reduction [16,17]	150% increase [22,23]
GFR (mL/min/1.73 m^2^)	90–120 [26]	Reduced, less than 90 [18]	160 (26 GW), 156 (36 GW) [8]

GW: gestational age in weeks.

**Table 2 pharmaceutics-15-02727-t002:** Physiological changes implemented in each cancer, pregnancy, and modified pregnancy population model relative to a non-pregnant healthy subject.

Physiological Parameter	Cancer (SimCYP V21)	Pregnancy (32 GW) (SimCYP V21)	Modified Pregnancy
Albumin	15% decrease	25% decrease	25% decrease
Alpha-1-acid glycoprotein (AGP)	100% increase	30% decrease	30% decrease
CYP3A4 abundance	No change	100% increase	100% increase
CYP2C8 abundance	No change	No change	150% increase ^#^
GFR (mL/min/1.73 m^2^)	30–40% decrease	25% increase	25% increase

Values are approximate estimates of the percent change of each physiological parameter value incorporated in the cancer population or pregnant population (at 32 gestational weeks) models relative to baseline values in the healthy volunteer population model (Simcyp^®^ V21). ^#^ A single modification on the abundance of hepatic CY2C8 activity (150% increase in the default value of 24 pmol CYP2C8 per mg microsomal protein) throughout pregnancy was made in the Simcyp^®^ pregnancy population model.

**Table 3 pharmaceutics-15-02727-t003:** Comparison of PK parameters of acalabrutinib between non-pregnant and pregnant patients with cancer.

Acalabrutinib 100 mg BID (Day 8)	C_max_ (ng/mL)	AUC_0–24h_ (ng/mL·h)
Observed: non-pregnant patients with cancer (Byrd et al. (2016) [38])	827	1850
Predicted: HV population	585	1702
Predicted: Cancer population	566	1719
Ratio HV/Cancer	1.03	0.99
Predicted: Pregnancy (32 GW)	251	662
Ratio Pregnancy/Cancer	0.44	0.39
Ratio Pregnancy/HV	0.43	0.39

Simulations representing healthy subjects, cancer, and pregnant populations were conducted using the “Sim-NEurCaucasian”, “Sim-Cancer” and “Sim-Pregnancy” population models, respectively. Ratio of C_max_ and AUC values predicted using each population model.

## Data Availability

Data are contained within the article and Appendix A.

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
