# Peer review of "Utility of Physiologically Based Pharmacokinetic Modeling to Investigate the Impact of Physiological Changes of Pregnancy and Cancer on Oncology Drug Pharmacokinetics"

_pharmaceutics, 2023, doi:10.3390/pharmaceutics15122727_

Round 1
Reviewer 1 Report
Comments and Suggestions for Authors
the manuscript by Yang et al addresses an important approach to overcome pharmacotherapeutic challenge in pregnant subjects with cancer. the manuscript is well-written and transparent. however there are few comments to improve the scientifc and technical aspects of the manuscript.
Introduction:
- Page 2: line 65-67 “… , and in vitro studies…” not all the cited references are in vitro studies.
- Table 1: reference 14 did not assess the change to CYP2C8 during pregnancy. Hence suggest removing “No change [14]” from this table. Changes to P-gp during pregnancy and in cancer populations was ignored in this table.
Material & Methods
- Page 3-Line 88: “animal data indicated no embryo-fetal toxicity or mortality….. “ you need to support this stamen with a reference at least.
- Page 4-line 125: paclitaxel fu 0.06, while the input parameters table says 0.054.
- Table 2: change to CYP3A4 abundance. It is very unlikely that an increase of 10% in CYP3A4 abundance can lead to any change in PK parameters during pregnancy. Was that supposed to be 150% or 130% as mentioned in line 170? Be consistent!
- Line 159-161: “the cancer model does not incorporate any changes in the expression of hepatic or liver CYP enzymes” was that supposed to read “… gut or liver CYP enzymes” ?
- How do you justify that the required changes to CYP2C8 to improve the paclitaxel model prediction during pregnancy and not the induction to the P-gp as this protein is induced by same mechanism as CYP3A4?
- How did you define the contributions of CYP3A5 expressors in each of docetaxel model predictions?
-
Results
- Add to figure 5b, the predicted mean profile without the 2-5-fold increase in CYP2C8.
Discussion
what are the advantages of using the developed PBPk model here over the recently published semi-physiological Pharmacokinetic Model for these drugs Janssen et al., (Clin Pharmacokinet. 2023 Aug;62(8):1157-1167).
How this model can be used for predicting PK in different gestational weeks?
It could be more useful if the authors shed light on the magnitude of dose modifications based on the developed model, whether a dose increase is required compared with non-pregnant subjects with cancer.
minors
- Figure 4: legends can be combined to improve the readability of the four plots.
Supplementary
Please add units for the input parameters in the tables, such as MW, ka, Qgut ..etc.
In all table there is a parameter called Kp scalar. What is this for ? maybe add a note under the table for these key parameter abbreviations,
Paclitaxel: what does the unit of P-gp CLint (../min/million ?) mean ? is it min/million hepatocytes ?
Docetaxel:
- Model development: “…. and additional hepatic metabolic clearance (HML=10%)” I guess I type here. Did you mean HLM rather than HML?
- Table S4: What does HAS stand for ?
- Pronk 2000 reported PK parameters from 14 subjects all received 75mg/m2 docetaxel. Why only 3 subjects from the smallest group were considered?
- Engels 2004 study: if the cmax and the clearance ratios were well-predicted, what was the reason for the underprediction of the AUC ratio? Did you consider the ratio of extrapolated AUC as it is the case for observed AUC-ratio?
- Table S6b: in this table you did not provide 90%CI of the ratio, while in other tables (Table S9b) you provided such quantity.
Acalabrutinib:
- Does “R” in CmaxR and AUCR stand for ratio ?
- Table S7: what is Vsac? Why few interaction parameters are with lower case “i” and few with uppercase “I”? Do they stand for different mechanisms or something else?
- Table S12: the %fm and %fe are missing. since CP3A4 changes during pregnancy, fm will be gestational time-dependent for both 3A4, 2C8 as well as for fe, which also affected by GFR changes in addition to CYP3A4. Suggest adding gestational ages for which these predictions were performed (if it is at 32 weeks as mentioned in page 6 of the main manuscript) and include fm at these gestational weeks. It would be more informative to add sample size to the observation and any variability in the observed parameters. This applied to Table S13 as well.
Table S14: second column: I think clarification would be appreciated. maybe say “Nonpregnant (with cancer)” instead of “Sim-cancer” and “Pregnant(with cancer)” instead of “Sim-Pregnancy”. Alternative give a title of “cancer” for that column and in the rows says non-pregnant or pregnant. You may have a better idea.
Author Response
Dear Sir/Ma’am,
The authors would like to express appreciation to the reviewers for their time and insightful comments. Please find responses to address the reviewers’ comments.
Reviewer 1
Comments and Suggestions for Authors
the manuscript by Yang et al addresses an important approach to overcome pharmacotherapeutic challenge in pregnant subjects with cancer. the manuscript is well-written and transparent. however there are few comments to improve the scientifc and technical aspects of the manuscript.
Introduction:
- Page 2: line 65-67 “… , and in vitro studies…” not all the cited references are in vitro studies.
Response: Clarified in the manuscript. References 14 and 18 are review papers that addressed CYP3A4. References 19 and 20 are the two in vitro studies with regards to CYP2C8.
- Table 1: reference 14 did not assess the change to CYP2C8 during pregnancy. Hence suggest removing “No change [14]” from this table. Changes to P-gp during pregnancy and in cancer populations was ignored in this table.
Response: Removed Reference 14 from Table 1 in the manuscript.
Material & Methods
- Page 3-Line 88: “animal data indicated no embryo-fetal toxicity or mortality….. “ you need to support this stamen with a reference at least.
Response: Clarified and reference 47 added in the manuscript.
- Page 4-line 125: paclitaxel fu 0.06, while the input parameters table says 0.054.
Response: Corrected to fu 0.05 in the manuscript.
- Table 2: change to CYP3A4 abundance. It is very unlikely that an increase of 10% in CYP3A4 abundance can lead to any change in PK parameters during pregnancy. Was that supposed to be 150% or 130% as mentioned in line 170? Be consistent!
Response: Corrected to 100% increase in the manuscript.
- Line 159-161: “the cancer model does not incorporate any changes in the expression of hepatic or liver CYP enzymes” was that supposed to read “… gut or liver CYP enzymes” ?
Response: Corrected “hepatic” to “gut” in the manuscript.
- How do you justify that the required changes to CYP2C8 to improve the paclitaxel model prediction during pregnancy and not the induction to the P-gp as this protein is induced by same mechanism as CYP3A4?
Response: We evaluated the pregnancy-induced changes in CYP2C8 instead of P-gp for the following reasons. First, in vitro data was available supporting induction of CYP2C8 by pregnancy related hormones and this induction was comparable in magnitude to that of CYP3A. Second, the contribution of CYP2C8 to paclitaxel clearance is considerably more significant than that of P-gp. Thus, it would be expected that changes in CYP2C8 would have a more significant impact on paclitaxel exposure than P-gp. Third, there are a lack of in vitro and limited in vivo data demonstrating changes in P-gp abundance/activity during pregnancy to support the assumption and potential magnitude of change.
- How did you define the contributions of CYP3A5 expressors in each of docetaxel model predictions?
Response: Based on evidence that a correlation exists between the abundances of CYP3A4 and CYP3A5 (PMID: 10086328; PMID: 12065767), the model incorporates a correlation equation, including a inter-subject variability, when calculating the abundance of CYP3A5 in each subject and subsequently, the percent contribution of CYP3A5 to docetaxel for each subject.
Results
- Add to figure 5b, the predicted mean profile without the 2-5-fold increase in CYP2C8.
Response: Added in the manuscript.
Discussion
what are the advantages of using the developed PBPk model here over the recently published semi-physiological Pharmacokinetic Model for these drugs Janssen et al., (Clin Pharmacokinet. 2023 Aug;62(8):1157-1167).
Response: Firstly, the authors examined alterations in CYP2C8 expression during pregnancy and validated the in vitro findings by incorporating the enhanced enzyme activity into the model. This modification in the system led to improved model predictions for paclitaxel. Secondly, the authors discussed the potential application for predicting the exposure of molecularly targeted oncology drugs in pregnant patients with cancer using the pregnancy population model, as demonstrated in the acalabrutinib case.
How this model can be used for predicting PK in different gestational weeks?
Response: The pregnancy model in Simcyp incorporates longitudinal changes in physiological parameters during pregnancy from week 1 to term. The predictive performance of this model for the first and second trimester of pregnancy for drugs that are CYP3A substrates has been demonstrated in other publications (e.g., reference 42). Our study focused on early third trimester based on availability of clinical PK data in pregnant patients with cancer used for model verification.
It could be more useful if the authors shed light on the magnitude of dose modifications based on the developed model, whether a dose increase is required compared with non-pregnant subjects with cancer.
Response: Thank you for the excellent comment. Although the views expressed in this paper are those of the authors and do not necessarily represent those of the US FDA, any recommendations of dosages/dosage modifications outside of the approved recommended labeling for the drug products may be miscontrued as the official position of the US FDA.
minors
- Figure 4: legends can be combined to improve the readability of the four plots.
Response: The 4 plots refer to different studies/references and the authors would respectfully ask to retain the current legends.
Supplementary
Please add units for the input parameters in the tables, such as MW, ka, Qgut ..etc.
Response: Input parameters have units in the supplementary.
In all table there is a parameter called Kp scalar. What is this for ? maybe add a note under the table for these key parameter abbreviations,
Response: Clarified in the supplementary.
Paclitaxel: what does the unit of P-gp CLint (../min/million ?) mean ? is it min/million hepatocytes?
Response: Clarified as uL/min/ million cells in the supplementary.
Docetaxel:
- Model development: “…. and additional hepatic metabolic clearance (HML=10%)” I guess I type here. Did you mean HLM rather than HML?
Response: Corrected in the supplementary to HLM.
- Table S4: What does HAS stand for ?
Response: Corrected in the supplementary to HSA (human serum albumin).
- Pronk 2000 reported PK parameters from 14 subjects all received 75mg/m2 docetaxel. Why only 3 subjects from the smallest group were considered?
Response: Modifications were made in Table S6a to include mean PK parameters from all 14 subjects who received docetaxel 75mg/m2.
- Engels 2004 study: if the cmax and the clearance ratios were well-predicted, what was the reason for the underprediction of the AUC ratio? Did you consider the ratio of extrapolated AUC as it is the case for observed AUC-ratio?
Response: Author reported AUCinf ratio from the simulation to compare with the value reported from the study (clarification is added into Table S6b). The overpredicted CL ratio does explain underpredicted AUC ratio.
- Table S6b: in this table you did not provide 90%CI of the ratio, while in other tables (Table S9b) you provided such quantity.
Response: 95% CI of the ratio (Engels 2004 study) were added to Table S6b. No CI were provided in the Engels 2006 study.
Acalabrutinib:
- Does “R” in CmaxR and AUCR stand for ratio ?
Response: Clarified in the supplementary.
- Table S7: what is Vsac? Why few interaction parameters are with lower case “i” and few with uppercase “I”? Do they stand for different mechanisms or something else?
Response: The definition for Vsac, Ki and KI is clarified in the manuscript. Vsac (L/Kg): Volume of single adjusting compartment; Ki, inhibitory constant for reversible inhibition; KI, inhibitory constant for time-dependent inhibition;
- Table S12: the %fm and %fe are missing. since CP3A4 changes during pregnancy, fm will be gestational time-dependent for both 3A4, 2C8 as well as for fe, which also affected by GFR changes in addition to CYP3A4. Suggest adding gestational ages for which these predictions were performed (if it is at 32 weeks as mentioned in page 6 of the main manuscript) and include fm at these gestational weeks. It would be more informative to add sample size to the observation and any variability in the observed parameters. This applied to Table S13 as well.
Response: The fm and fe information were added to Table S12. Sample size and gestational weeks were provided in Tables S12 and S13. However, as the observed values were digitized from the reference, no variability is provided.
Table S14: second column: I think clarification would be appreciated. maybe say “Nonpregnant (with cancer)” instead of “Sim-cancer” and “Pregnant(with cancer)” instead of “Sim-Pregnancy”. Alternative give a title of “cancer” for that column and in the rows says non-pregnant or pregnant. You may have a better idea.
Response: Clarified in the supplementary.
Reviewer 2
Comments and Suggestions for Authors
The authors address an interesting question about the possible reasons why there may be differences in the exposure of pregnant vs nonpregnant patients. However, I am not entirely convinced that the author's approach is getting at some fundamentally useful information that can help improve cancer outcomes of pregnant patients, especially if it comes at (an unecessary?) expense of affecting the life or the development of the baby or increased chances that perhaps the best decision is for the mother to deliver the baby before undergoing chemo. After all, perhaps the pregnancy could come to completion before any treatment is initiated? At what point is chemotherapy useful in pregnancy? Could it be better to wait to begin treatment at the 'right time' before going through the guesswork that the models suggest?
Response: Thank you for your insightful questions. The purpose of our research is to investigate the effects of pregnancy and cancer on the PK of oncology drugs and communcate the findings to potentially inform dosing for pregnant patients with cancer. The management of cancer and pharmacotherapeutic benefits vs. risks to the mother and fetus of continued cancer drug use during pregnancy should be considered individually for the pregnant patient based on the patient, oncologist’s judgement, and patient’s healthcare team. A sentence is added in the Introduction.
Also, in the nonpregnant cancer model, what kind of cancer is being treated and in what kind of population is this being modeled? Are there different paclitaxel treatment regimens recommended for different kinds of cancer and is this relevant in terms of whether this model is generalizable to all kinds of paclitaxel treatable cancers? Is paclitaxel or docetaxel or Drug X used alone or in combination in the cancer patients, and is there a possibility that a drug combination issue may be something to deal with? What are the ultimate effects on health care outcomes (e.g. morbidity/mortality) associated with neutropenia/cytopenia in pregnant women with cancer vs regular cancer patients?What is the ultimate efficacy of paclitaxel against tumors in a pregnant women, given that those tumors are going to be experiencing a completely different hormonal environment and thus may be under completely different physiological growing conditions? Perhaps it is just better not to treat during pregnancy, given how little we know? Are ther alternative treatments that could be better predicted with a PBPK model and that perhaps do not pose as much a risk to mother or baby (e.g. monoclonal antibodies)? How much do we really lose if a 7month pregnant woman undergoes a C-section before being treated? Is paclitaxel/docetaxel treatment involve a bone marrow transplant? Can this bone marrow transplant be done on a pregnant woman? Please pardon my ignorance that I have to ask all these questions, but I think that these are the kinds of questions that we need to be asking - and those questions may or may not be answered by doing an experiment guided by a PBPK model -but perhaps they can. I am just not sure at this point.
Response: Thank you for your insightful questions. There are different paclitaxel treatment regimens for different cancers. We chose to evaluate paclitaxel 175 mg/m2 every 3 weeks as this is the appeoved recommended dosage for breast cancer, which is the most common cancer in pregnancy. Health care outcomes (e.g. morbidity/mortality) in pregnant women with cancer vs. non-pregnant cancer patients and efficacy of paclitaxel in pregnant women are outside the scope of the manuscript and research question.
Thank you for your great question on monoclonal antibodies. The placental transfer of monoclonal antibodies can occur via a specific receptor-mediated mechanism from gestational week 16 onward and may not cause any major fetal malformations in the first trimester. On the other hand, small-molecule agents are likely to cross the placenta during the first trimester and may cause fetal malformation and increase the rate of spontaneous miscarriage. Further research on PBPK modeling of biological products for pregnant patients with cancer are warranted.
Reviewer 2 Report
Comments and Suggestions for Authors
The authors address an interesting question about the possible reasons why there may be differences in the exposure of pregnant vs nonpregnant patients. However, I am not entirely convinced that the author's approach is getting at some fundamentally useful information that can help improve cancer outcomes of pregnant patients, especially if it comes at (an unecessary?) expense of affecting the life or the development of the baby or increased chances that perhaps the best decision is for the mother to deliver the baby before undergoing chemo. After all, perhaps the pregnancy could come to completion before any treatment is initiated? At what point is chemotherapy useful in pregnancy? Could it be better to wait to begin treatment at the 'right time' before going through the guesswork that the models suggest?
Also, in the nonpregnant cancer model, what kind of cancer is being treated and in what kind of population is this being modeled? Are there different paclitaxel treatment regimens recommended for different kinds of cancer and is this relevant in terms of whether this model is generalizable to all kinds of paclitaxel treatable cancers? Is paclitaxel or docetaxel or Drug X used alone or in combination in the cancer patients, and is there a possibility that a drug combination issue may be something to deal with? What are the ultimate effects on health care outcomes (e.g. morbidity/mortality) associated with neutropenia/cytopenia in pregnant women with cancer vs regular cancer patients?What is the ultimate efficacy of paclitaxel against tumors in a pregnant women, given that those tumors are going to be experiencing a completely different hormonal environment and thus may be under completely different physiological growing conditions? Perhaps it is just better not to treat during pregnancy, given how little we know? Are ther alternative treatments that could be better predicted with a PBPK model and that perhaps do not pose as much a risk to mother or baby (e.g. monoclonal antibodies)? How much do we really lose if a 7month pregnant woman undergoes a C-section before being treated? Is paclitaxel/docetaxel treatment involve a bone marrow transplant? Can this bone marrow transplant be done on a pregnant woman? Please pardon my ignorance that I have to ask all these questions, but I think that these are the kinds of questions that we need to be asking - and those questions may or may not be answered by doing an experiment guided by a PBPK model -but perhaps they can. I am just not sure at this point.
Author Response
Dear Sir/Ma’am,
The authors would like to express appreciation to the reviewers for their time and insightful comments. Please find responses to address the reviewers’ comments.
Reviewer 2
Comments and Suggestions for Authors
The authors address an interesting question about the possible reasons why there may be differences in the exposure of pregnant vs nonpregnant patients. However, I am not entirely convinced that the author's approach is getting at some fundamentally useful information that can help improve cancer outcomes of pregnant patients, especially if it comes at (an unecessary?) expense of affecting the life or the development of the baby or increased chances that perhaps the best decision is for the mother to deliver the baby before undergoing chemo. After all, perhaps the pregnancy could come to completion before any treatment is initiated? At what point is chemotherapy useful in pregnancy? Could it be better to wait to begin treatment at the 'right time' before going through the guesswork that the models suggest?
Response: Thank you for your insightful questions. The purpose of our research is to investigate the effects of pregnancy and cancer on the PK of oncology drugs and communcate the findings to potentially inform dosing for pregnant patients with cancer. The management of cancer and pharmacotherapeutic benefits vs. risks to the mother and fetus of continued cancer drug use during pregnancy should be considered individually for the pregnant patient based on the patient, oncologist’s judgement, and patient’s healthcare team. A sentence is added in the Introduction.
Also, in the nonpregnant cancer model, what kind of cancer is being treated and in what kind of population is this being modeled? Are there different paclitaxel treatment regimens recommended for different kinds of cancer and is this relevant in terms of whether this model is generalizable to all kinds of paclitaxel treatable cancers? Is paclitaxel or docetaxel or Drug X used alone or in combination in the cancer patients, and is there a possibility that a drug combination issue may be something to deal with? What are the ultimate effects on health care outcomes (e.g. morbidity/mortality) associated with neutropenia/cytopenia in pregnant women with cancer vs regular cancer patients?What is the ultimate efficacy of paclitaxel against tumors in a pregnant women, given that those tumors are going to be experiencing a completely different hormonal environment and thus may be under completely different physiological growing conditions? Perhaps it is just better not to treat during pregnancy, given how little we know? Are ther alternative treatments that could be better predicted with a PBPK model and that perhaps do not pose as much a risk to mother or baby (e.g. monoclonal antibodies)? How much do we really lose if a 7month pregnant woman undergoes a C-section before being treated? Is paclitaxel/docetaxel treatment involve a bone marrow transplant? Can this bone marrow transplant be done on a pregnant woman? Please pardon my ignorance that I have to ask all these questions, but I think that these are the kinds of questions that we need to be asking - and those questions may or may not be answered by doing an experiment guided by a PBPK model -but perhaps they can. I am just not sure at this point.
Response: Thank you for your insightful questions. There are different paclitaxel treatment regimens for different cancers. We chose to evaluate paclitaxel 175 mg/m2 every 3 weeks as this is the appeoved recommended dosage for breast cancer, which is the most common cancer in pregnancy. Health care outcomes (e.g. morbidity/mortality) in pregnant women with cancer vs. non-pregnant cancer patients and efficacy of paclitaxel in pregnant women are outside the scope of the manuscript and research question.
Thank you for your great question on monoclonal antibodies. The placental transfer of monoclonal antibodies can occur via a specific receptor-mediated mechanism from gestational week 16 onward and may not cause any major fetal malformations in the first trimester. On the other hand, small-molecule agents are likely to cross the placenta during the first trimester and may cause fetal malformation and increase the rate of spontaneous miscarriage. Further research on PBPK modeling of biological products for pregnant patients with cancer are warranted.

Round 2
Reviewer 2 Report
Comments and Suggestions for Authors
The authors have not done any additional work to improve the manuscript in response to my comments. As it stands, the work that the authors have done is a simple PBPK modeling exercise, without considering the serious implications and complications that may arise to mother and fetus of treating cancer in pregnancy without considering all the different variables that accompanies pregnancy, and the alternative treatment options. Based on what they write and how they respond, I would say that the medically sound recommendation would be to perform C-section (or abort the fetus if pregnancy is early) to save the mother, as disregarding the specific complications that arise from pregnancy in terms of treatment outcomes could increase the probability that both the mother and the fetus will die. If the authors had been dealing with some other kind of drug that would not pose such a life-death situation, or if the authors would have seriously considered my criticisms in regards to the decision of whether to treat, or abort the baby, or wait until a C-section and then treat the mother, then perhaps I would be more open towards considering the results of their PBPK modeling study. I am not a bioethicist, but I think there are ethical implications in publishing this kind of study and the authors may want to consult with a bioethicist if they are planning to submit this work for publication elsewhere.